# Workplace Interventions to Reduce Occupational Stress for Older Workers: A Systematic Review

**DOI:** 10.3390/ijerph19159202

**Published:** 2022-07-27

**Authors:** Daniel Subel, David Blane, Jessica Sheringham

**Affiliations:** 1Institute of Epidemiology and Health Care, University College London, London WC1E 6BT, UK; 2Faculty of Medicine, School of Public Health, Imperial College London, London SW7 2AZ, UK; d.blane@imperial.ac.uk; 3Department of Applied Health Research, University College London, London WC1E 6BT, UK; j.sheringham@ucl.ac.uk

**Keywords:** intervention, workplace, occupational stress, older workers

## Abstract

The working life of individuals is now longer because of increases to state pension age in the United Kingdom. Older workers may be at particular risk in the workplace, compared with younger workers. Successful workplace interventions to reduce occupational stress amongst older workers are essential, but little is known about their effectiveness. The aim is to evaluate current evidence of the effectiveness of interventions for reducing stress in older workers in non-healthcare settings. Four database searches were conducted. The search terms included synonyms of “intervention”, “workplace” and “occupational stress” to identify original studies published since 2011. Dual screening was conducted on the sample to identify studies which met the inclusion criteria. The RoB 2.0 tool for RCTs was used to assess the risk of bias. From 3708 papers retrieved, ten eligible papers were identified. Seven of the papers’ interventions were deemed effective in reducing workplace stress. The sample size for most studies was small, and the effectiveness of interventions were more likely to be reported when studies used self-report measures, rather than biological measures. This review indicates that workplace interventions might be effective for reducing stress in older workers. However, there remains an absence of high-quality evidence in this field.

## 1. Introduction

By 2040, it is predicted that one in every two people of working age will be aged 65 or over [1]. The global ageing population has resulted in government concerns regarding the future of the workplace [2]. The increase in life expectancy and the lack of equitable social resources available has been a catalyst for most European governments to increase the state pension age [3].

Prolonging the working life of individuals cannot be done without due diligence and needs to be medically supervised, as suggested by MISPA (Mitigating Increases in the State Pension Age) [4]. Before governments can continue increasing state pension age, it needs to be assessed how this can be conducted, without damaging or harming the health of workers affected by these changes–particularly workers in physically demanding and highly stressful occupations.

Older workers face greater or different hazards to their health than younger workers. Bravo et al.’s [5] review found that in 50% of the papers they reviewed, older workers were at a much greater risk of fatal workplace injuries, when compared to their younger counterparts. Older workers are more likely to have pre-existing long-term health conditions, which can affect their capacity to work or the kinds of work they are able to sustain. There is also evidence that they experience greater sickness absence [6].

Stress—adverse reactions to excessive pressure—and burnout are recognised as a major risk to the health of all workers [7,8]. There is conflicting evidence whether older workers are at a greater risk of stress than younger workers [9]. It is clear, however, that older workers are likely to face different stressors to younger workers, not just through pressures within the workplace, but also through additional caring responsibilities outside of work [4]. Moreover, there is agreement that, despite legislation to prevent it, there is evidence that older workers are subject to age discrimination [10]. Therefore, interventions to improve workplace health in older workers may well need to be different to those of younger workers because it cannot be assumed that the problems they face, or the mechanisms by which interventions work, will be the same.

Based on currently available research, very little is known about this topic. Evidence of the effectiveness of workplace interventions for older workers is lacking. Poscia et al.’s [3] systematic review found a paucity of high-quality evidence on workplace health promotion for older workers. There was a suggestion that active workplace interventions help improve the health of older workers, but included studies used small, convenience samples not representative of the working population.

Pieper et al.’s [11] more recent review of reviews on workplace health promotion interventions found that psychological interventions, such as stress management, cognitive behavioural therapy and mindfulness-based interventions have the ability to significantly reduce stress. However, Pieper et al. found few reviews specifically focusing on older workers and they reported that there was insufficient evidence to conclude that psychological interventions were the most successful and effective to reduce occupational stress amongst older workers. Interventions were predominately targeted towards white-collar workers, teachers, and healthcare providers. Interventions for healthcare providers may be of limited generalisability to other settings, given the specific nature of healthcare settings and healthcare work and the hazards that may present in this environment. When assessing previous studies on this topic, the overwhelming majority focus upon younger people employed in advantaged occupations, using small cohort sizes. Furthermore, they use inconsistent and haphazard outcome measures to assess interventions’ successes, which results in studies being unrepresentative, difficult to replicate and unable to demonstrate the impact on increased state pension age for older workers.

Before policy makers can enact changes to state pension age, they must have access to a sufficient level of high-quality research which has outlined the impact on individuals working longer, as well as interventions used to retain older workers. This information must also be accessible to employers, so they are made aware of the most successful interventions in the workplace to reduce occupational stress and maintain their workforce. This article intends to provide policy makers and employers a review of the current literature and research in this field. This study also has the potential to provide union representatives, and workers themselves, with evidence for them to vouch to their employers for adequate, appropriate, and successful interventions in their workplaces.

This systematic review sought to answer the question:

“What is the evidence of effectiveness of workplace interventions for reducing occupational stress in older workers outside of the healthcare sector?”

The objectives were to:(1)Identify and appraise papers evaluating the effectiveness of workplace stress reduction interventions on older workers.(2)Describe the types of interventions and measures of effectiveness used(3)Summarise the evidence of effectiveness of interventions(4)Identify existing knowledge gaps in the literature which require further research

## 2. Materials and Methods

### 2.1. Search Strategy

PRISMA guidelines for reporting systematic reviews were followed throughout the process of this review [12], see checklist in Appendix A. Four database searches were conducted: OvidMedline, PsycInfo, Scopus and Web of Science. After an initial literature search, a PICO model was developed (see Table 1), which helped form the database search terms for the review [13]. Previously systematic reviews’ search terms, including Pieper et al. [11] and Poscia et al. [3], helped to inform the search terms. The search term combinations were first applied in OvidMedline, which uses MeSH terms, and then modified and adapted for use in the other databases (see Table 2). In all the databases, the presence of key words was sought in “all fields”, which would detect the terms in key words, titles, abstracts and full papers. Initially age terms were included in the search strategy, but this resulted in an improbably low number of results retrieved, so this term was dropped. The searches took place throughout the first week of August 2021, therefore only research published before 31 July 2021 were included in this review.

### 2.2. Inclusion/Exclusion Criteria

Table 3 depicts the inclusion/exclusion criteria. Eligible papers had to report an intervention in a non-health sector workplace, specifically focusing on older workers. The papers had to have been conducted in an Organisation for Economic Co-operation and Development (OECD) country, to ensure findings have some relevance to the United Kingdom (UK) context [14]. Studies without a control group or baseline data, or without an aim of reducing workplace stress, were excluded. The authors did not set out to select papers which specified a specific control condition but sought papers which described what interventions were compared with. Qualitative papers, such as focus groups or interviews, were excluded from this review as quantitative papers were deemed to illustrate more objective results and are more likely to be conducted on a large number of participants.

The definition of an older worker was developed by adapting multiple definitions from various sources. Firstly, if the paper classified the intervention or participants as older workers, regardless of the mean age, these interventions were deemed to be focused on older workers. Secondly, for OECD countries, the average age at which an individual reached normal pension age in 2016 was 63.7 years old for women and 64.3 years old for men [15]. If the mean age of participants in a paper were within 15 years of normal pension age, it was concluded that older workers were included in this intervention.

### 2.3. Study Selection and Screening

Papers from the four databases were exported to Microsoft Excel. Title and abstracts of all papers screened by DS (author and reviewer) and a secondary reviewer (AH). Any papers which were unclear or resulted in polarized views, were then resolved by discussion with a third reviewer and co-author (JS). After the title and abstract screening, the remaining papers underwent a full-text screening.

Each paper that met the inclusion criteria on screening was carefully assessed for its relevance to older workers. Papers that were specifically focused on older workers were placed in the primary dataset. Papers where data on the effectiveness of the intervention on older workers was included, but without a specific focus on older workers, comprised the secondary dataset.

### 2.4. Data Extraction and Critical Appraisal

Data were extracted from all eligible papers used a data extraction form by DS, with a sample checked by JS (see Appendix B: Data Extraction Form) to cover features including: study design and employment setting; the age of participants; nature of the intervention; reported effectiveness. Interventions were coded into three categories–psychological interventions, educational interventions, and physical interventions. Outcome measurements were grouped by whether self-report or biological samples were used to measure stress.

The RoB 2.0 tool (Risk of bias in randomised trials) [16] was used by DS and JS in each paper to form a judgement about the risk of bias across six different domains. The RoB 2.0 tool was chosen as it enabled the reviewers to form their own assessment of an article’s quality, in regard to its risk of different types of biases. If a domain or the overall judgement was deemed to have a high risk of bias, this meant that the reviewers believed that there was an issue with the paper that substantially lowered their confidence in the results. Some concerns of bias indicated that a paper included an issue which could potentially lower the reviewer’s confidence in the results. If the overall judgement was that the paper had a low risk of bias, this meant that the reviewers were confident that the study results were valid.

The included studies were described, and the characteristics and methods for ascertaining stress levels were summarised. Based upon what was written in each paper, the effectiveness of the interventions was summarised, using quantitative data to assess the success of each intervention.

## 3. Results

### 3.1. Characteristics of Included Studies

From 3708 papers identified in the database searches, ten papers met the inclusion criteria (Figure 1). Five papers had a specific focus on older workers (the primary dataset). A further five papers did not have a specific focus of the research on older workers (the secondary dataset). As the mean age of participants in both datasets were similar (see Table 4), they are considered as one dataset in the rest of the paper.

Five papers were conducted in the United States [17,18,19,20,21]; the other five originated from Europe (Germany [22], the Netherlands [23], Finland [24], Italy [25] and Norway [26]). The number of participants ranged from 14–779, with three studies have less than 40 participants. Only one study included over 500 participants [24] (see Table 4).ijerph-19-09202-t004_Table 4Table 4Description of Studies.Study (First Author, Year)CountryStudy DesignFocus on Older Workers?Participant’s OccupationAge of ParticipantsNo. of Participants (and Dropouts)Primary DatasetHughes, 2011 [17]United StatesRandomised Controlled Trial (RCT)YesUniversity Staff51 (Mean)423(56 Dropouts)Malarkey, 2013 [18]United StatesRandomised Controlled TrialYesUniversity Faculty Staff50 (Mean)186(0 Dropouts)Cook, 2015 [20]United StatesRandomised Controlled TrialYesTech Company Workers59 (Median) *50–68 (Range)278(0 Dropouts)Fischetti, 2019 [25]ItalyRandomised Controlled TrialYesPolice Officers46.8 (Mean)20(0 Dropouts)Calogiuri, 2016 [26]NorwayRandomised Controlled TrialYesOffice Workers49 (Median) *41–47 (Range)14(3 Dropouts)Secondary DatasetAikens, 2014 [19]United StatesRandomised Controlled TrialNoChemical Company Employees41.5 (Median) *18–65 (Range)89(23 Dropouts)Largo-Wight, 2017 [21]United StatesRandomised Controlled TrialNoUniversity Office Staff48.8 (Mean)37(0 Dropouts)Limm, 2011 [22]GermanyRandomised Controlled TrialNoLower and Middle LevelManagers at a Manufacturing Plant40.9 (Mean)18–65 (Range)174(20 Dropouts)Hoeve, 2021 [23]NetherlandsQuasi-ExperimentalNoPolice Officers49 (Mean)30–63 (Range)82(19 Dropouts)Ojala, 2019 [24]FinlandNon-Randomised TrialNoPublic Sector Workers *49.9 (Mean)21–64 (Range)779(217 Dropouts)* Median has been calculated by the researcher as the midpoint between the range. In Ojala’s study, Public Sector. workers included construction and transport workers, office workers, food services and managerial specialists.

Three studies were conducted with university faculty staff [17,18,21], and three in manufacturing or technical environments [19,20,22]. Two studies were conducted amongst police officers [23,25]. The remaining two studies were conducted with office workers [24,26].

The age of participants was described in two ways (Table 4). Six studies described the age range of participants in the intervention; the upper limit for the age range was between 57 and 68; the lower limit for the age range was 18 to 50. Three papers only included participants over the age of 40, with Calogiuri et al.’s [26] paper using only participants older than 50 years old. In the seven papers that documented the mean age of participants, mean age was over 40.9 years. Five papers had a mean age of over 48 years [17,18,21,23,24].

### 3.2. Risk of Bias

None of the papers had an overall high risk of bias (Table 5). Four papers were judged to have a low risk of bias. Some bias concerns were identified in six papers. In nine out of the ten papers there was a lack of detail on the randomisation of participants, which may have led to post-test reporting bias by participants exaggerating the effects of the intervention. Most papers showed a strong adherence to the intended intervention. Fischetti et al.’s [25] study showed a potential high risk of bias in the measurement of outcome. While the study used validated scales to assess stress, the score was high because of the study’s pre-post evaluation design. It is possible that participants may be subject to bias in overestimating the effects of participation on their well-being.

### 3.3. Study Methods

The most common form of intervention was psychological interventions (*n* = 8). Psychological interventions included mindfulness-based, cognitive behavioural therapy and stress management interventions. Three studies used physical interventions, which involved exercise, walking, weight training or circuit training programmes [20,25,26]. One paper included an educational intervention [17] focused on health education (Table 6).

Of the ten papers in this review, five papers [19,20,22,24,25] reported that the control group received no intervention during the research but were waitlisted to participate in the intervention at a later date. In Malarkey et al.’s [18] study, the participants in the control group received a lifestyle and educational intervention, compared with the mindfulness-based intervention that the experimental group received. Hughes et al.’s [17] study control group received a light level of health education compared with the experimental group, who received the health promotion intervention. Hoeve et al.’s [23] control group received a regular education intervention, without any mindfulness training. Two papers’ control groups [21,26] had either an indoor standard work break or indoor exercise, compared to the experimental groups whose interventions were conducted outdoors. No conclusive pattern emerged between which control condition was in place and the outcome of the intervention. Table 6 illustrates that of the five interventions [19,20,22,24,25] in which the control group received no intervention, three of these papers reported an effective intervention in the experimental group.

Six papers conducted their interventions in the workplace offices, two papers were carried out via online means in the workplace, and a further two papers took place outside of the workplace, in green areas and nature. 

All papers in this review used self-reported questionnaires to collect data on stress. Three of these papers also collected cortisol levels, either from saliva samples or blood tests [18,22,26]. Four of the papers used the Perceived Stress Scale Questionnaire to assess the level of psychological stress perceived in participants. 

The shortest intervention took place over the course of two weeks [26]. Three of the papers’ interventions took place for over six months, including follow up time [17,22,24]. The longest duration for intervention was Hughes et al.’s [17] 12-month study.

### 3.4. Study Findings

Changes in stress levels as a result of each intervention are reported in Table 6. In seven out of the ten studies [19,21,22,23,24,25,26], there was improvement in at least one measure of self-reported stress levels. However, none of the three studies that measured cortisol levels [18,22,26] found any significant differences between the intervention and the control group’s cortisol levels. 

Three interventions [17,18,20] showed no evidence of effectiveness on any measure. There were no consistent patterns in terms of the intervention type (psychological, physical educational), workplace setting or delivery method between effective and ineffective interventions.

## 4. Discussion

### 4.1. Main Findings

The evidence of the effectiveness of interventions to reduce stress in older workers was varied. Seven out of the ten papers reported some effectiveness in reducing self-reported stress in older workers as a result of interventions. Studies that measured cortisol levels did not report any reduction in stress. Most of the interventions were psychological in nature, but there was no difference in reported effectiveness between psychological, physical, or educational interventions. It should be noted that most of these interventions were only short-term, and therefore, longer-term impacts of these interventions are not clearly demonstrated.

### 4.2. Methodological Considerations

There were some important limitations in the studies included in the review. Firstly, the number of participants reported in each study was generally low, with three papers comprising studies of less than 40 participants and only one study with more than 500. Due to the low number of participants, it is difficult to generalise the results of these interventions to the broader population [27]. In most of the studies, participants had to volunteer to take part. In some studies, it was not clear how many employees that were eligible declined to take part so the acceptability of such interventions in the workplace cannot be concluded from this study.

Secondly, none of the ten papers observed the longer-term impacts of the interventions. Whilst papers stated or implied that the interventions were longer-term solutions to the problem of occupational stress amongst older workers, they provide no conclusive evidence of long-term benefit. The concern regarding the long-term effects associated with workplace interventions has also been discussed by others. Steenstra et al. [28] reported how the effect of interventions require a very long follow-up, which is extremely difficult to achieve and maintain. They concluded that the interventions’ effect would most likely dilute over time and not result in any long-term benefits. Similarly, in this review, two out of the three papers which had interventions lasting more than 8 months were shown to have mixed or no effect on reducing workplace stress. This is suggestive evidence in support of Steenstra et al.’s conclusions that the impact of workplace interventions to reduce stress could have little to no long-term benefit if the intervention is not maintained in the workplace. It is also possible maintenance of a short-term intervention is not enough; workplaces may need different kinds of approaches to maintain reductions in stress levels in the longer term.

Thirdly, there was a range of self-report questionnaires used, which collected data on various aspects relating to stress, mental health, or other factors. When analysing the interventions, as different measurement outcomes are used, it can cause difficulties in understanding which intervention is the most effective. 

There were some limitations also in the conduct of this review. Only papers published in English were included in the review. Studies which were written and published in other languages were removed at the first stage of screening. Whilst the majority of papers which were found in the database search stage of the review were written in English, those in other languages may have been beneficial to include in this review. Using free, online translation software to translate any non-English studies has become more common in academic reviews, and if this research was to be conducted again in the future, including non-English studies, and using translation software should be strongly considered. It was also beyond the scope of the review to include qualitative studies. Whilst these would not have definitely addressed questions of effectiveness, they could have provided useful insights into why intervention achieved their effects. The RoB 2.0 tool which was used for performing the critical appraisal does not prompt consideration of wider aspects of quality and relevance, for example, what the control conditions were. This could be seen as a potential limitation in several of the papers in the review.

A further challenge faced in this review was the ambiguity regarding the definition of an older workers. The initial search terms included specific terms and synonyms for “older worker”. However, this resulted in a very small number of papers being retrieved. Therefore, this search term was removed and at abstract and full paper screening, the reviewers determined which papers focused on older workers and which did not. Eliminating “older workers” as a search term in the database search led to a potential risk that relevant literature, with a clear focus on older workers, may have been overlooked. However, “older worker” was hard to define partly because the classification of an older worker varies across countries. The ELSA (English Longitudinal Study of Ageing) and the JSTAR (Japanese Study of Ageing and Retirement) both describe workers over the age of 50 as “older workers”, however, Kingston and Jagger [29] argue that cohort studies with the lower age limit of 50 to 65 years old, often have fewer very old people in the studies, therefore, are not fully representative of older workers. The nature of the risk of being an older worker varies in the context of workplace settings, occupations and job demands. For example, as seen in Fischetti et al.’s [25] and Hoeve et al.’s [23] research, police officers may be more at risk of injury at a younger age, due to the physical nature of their occupation. This may result in police officers aged 40 being deemed as older workers in their profession, although at their chronological age, in society and in other professional groups, they would be classified as younger. However, at this age, it is possible for some police officers that the nature of their work may change, to become more ‘desk based’. In this case, the current workplace exposures may be more similar to office workers, but the prior exposures they faced from working in communities may have long lasting and distinct effects on their health that are not experienced by those who have spent their entire careers in office-based jobs. Due to the small number of studies identified, this review was not able to explore the differences in the nature of interventions across workplaces. This is needed in future because different causes of stress based upon a range of diverse types of employment may affect the sustainability and the effectiveness of interventions to reduce workplace stress.

### 4.3. Interpretation of Findings and Comparison to Previous Studies

Of the ten papers sourced for this review, only five reported a specific focus on older workers, demonstrating the lack of robust and available literature on this topic. This finding is consistent with older systematic reviews researching workplace interventions for older workers [3,11,28]. More than ten years ago, Crawford et al. [30] urged for more research to be conducted on health and safety management interventions for workers over the age of 50 in relation to the physical and psychological changes that occur when workers reach this age. 

Interventions that used self-reported measures appeared more effective when compared to biological measures. However, it is important to note that self-reports and biological samples measure different things. Taking part in an intervention may improve subjective well-being in an individual, even if it has no biological effect. This does not imply that the intervention was unsuccessful or ineffective. Indeed, McDonald [31] suggests that gaining self-reported data from participants is the most logical way to learn more about an individual. Arguably, an individual’s subjective well-being is what would keep them in the workplace. 

Whilst it is understood that older workers might not always face more workplace stress compared to younger workers, they could be more at risk of specific stressors connected to responsibilities outside of work, age discrimination and physical health conditions that are more common in older age [32]. In the ten papers’ interventions, there was not enough description regarding the extent to which specific stressors associated with older age were addressed. It is, however, significant that there were five studies that did not seek to focus on older workers, potentially overlooking distinct stressors. In these studies, it is also possible that the overall effectiveness could have been driven by higher effectiveness in younger populations but there was not sufficient data reported in these papers on effects by participant age to explore this. The context in which the interventions effect change may be important. Interventions in the workplace, which are promoted and supported by employers may encourage participants not only to take part in the intervention, but also to make changes to their lifestyle and behaviour, which, in turn, would ultimately improve their well-being and decrease their stress levels at work [33].

### 4.4. The Significance of the Review and Public Health Implications

Since Crawford et al.’s [30] review was undertaken, policy and demographic changes have lead to a higher proportion of older workers in many countries, increasing the importance of health and safety interventions for workers over the age of 50. The need for such research has not been addressed and the knowledge gaps that were present in the literature remain. 

This review has demonstrated that there is still not sufficient research available for governments and policy makers to make an informed decision on the impact of increasing state pension age on the population. If they are determined to extend the working life of individuals, governments will need to ensure that there is no detriment to the health of older workers. 

The lack of high-quality literature on this topic results in this review being unable to provide any definitive conclusions regarding the most effective and successful workplace intervention to deal with occupational stress. This systematic review can be updated to illustrate newly published literature about older workers’ well-being in years to come. The significance of implementing a successful intervention to promote and maintain the health of older workers is vital for the longer-term wealth creation and sustaining of both the economy and health of the population [34]. 

This review has not shown an adequate amount of successful workplace interventions to support older workers’ occupational stress to mitigate the public health implications of raising state pension age, as reported by MISPA [4]. More extensive and robust research is required to illustrate to both employers and policy makers that increasing state pension age will result in; increased morbidity and mortality rates for those in demanding occupations; overwhelming the already sparse healthcare services-both for occupational health and primary care; and, worsening the health for workers who are already ill. Careful considerations need to be made to ensure that older workers are not adversely harmed by increases to state pension age. It is fundamental that interventions, which have been proven successful for older workers, must be introduced into more workplaces to ensure a smoother transition for older workers who are now working longer.

## 5. Conclusions

As the population ages, and statutory pension age increase, the proportion of older workers will increase in the workplace. Older workers face distinct and sometimes greater risks to health and well-being compared with younger workers, which may place them at particular risk of stress. This review found some promising evidence that interventions in the workplace can improve self-reported stress in older workers in the short term. It also highlighted the paucity of studies with interventions specifically designed for older workers. Further studies are required to understand longer term impacts of workplace interventions on older workers and to elucidate what type of intervention is most likely to be effective in different workplace settings.

## Figures and Tables

**Figure 1 ijerph-19-09202-f001:**
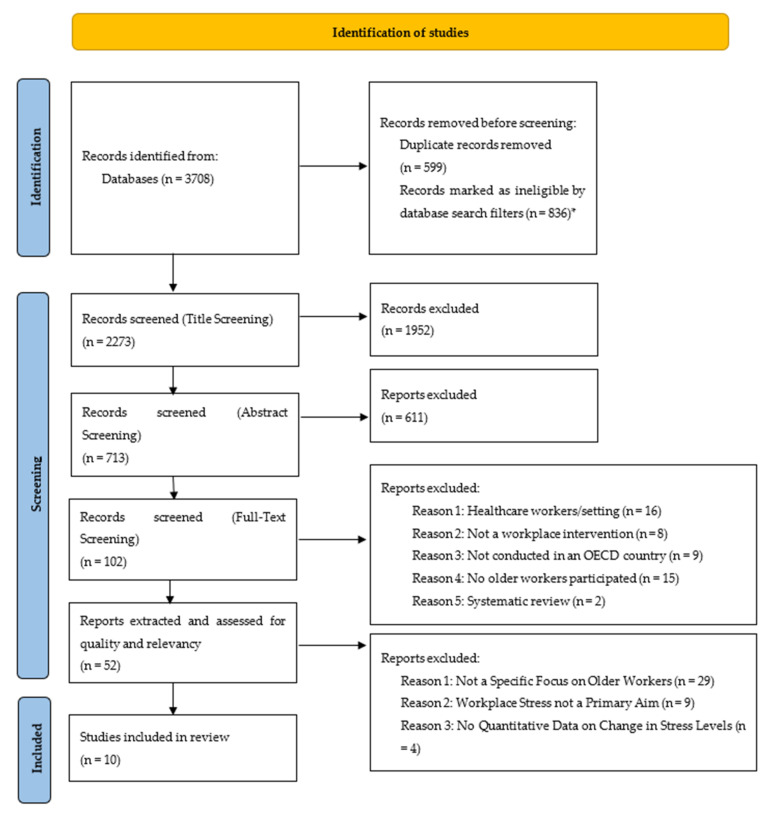
PRISMA Flow Chart (Adapted from Page et al. [12]) * Papers were automatically excluded using filters in the search databases where they were outside of our date range and language of publication.

**Table 1 ijerph-19-09202-t001:** The PICO model.

PICO Term	Detail
Population	Older workers, in the Organisation for Economic Co-Operation and Development (OECD) country, in non-health sector jobs.
Intervention	All interventions occurring in the workplace, including medication, educational and exercise interventions.
Control	Comparison with control conditions as described in each of the papers in the review
Outcome	Reduced workplace stress

**Table 2 ijerph-19-09202-t002:** Search Terms.

Database	Programmes Search Terms	Setting Search Terms	Outcome Search Terms	Papers per Database
OvidMedline	Intervention.mp. OR Psychosocial Intervention/	Workplace/ OR Workplace.mp.	Burnout, Professional/OR	2444
			Occupational Stress.mp. OR	
			Stress, Psychological/OR Occupational Stress/OR Occupational Diseases	
Scopus	Intervention OR Programme OR Program	Workplace OR Office OR “Work Centre”	“Occupational Stress” OR “Professional Burnout” OR “Psychological Stress”	701
Web of Science	Intervention OR Programmes OR Program	Workplace * OR Office * OR “Work Centre”	“Occupational Stress” OR “Professional Burnout” OR “Workplace Stress” OR “Job Stress” OR “Psychological Stress”	964
PsycInfo	Exp Workplace Intervention/OR Intervention.mp. OR exp intervention	Exp Workplace Intervention/OR Workplace.mp.	Occupational Stress.mp. OR exp Occupational Stress/	469
Total Papers		3708 Papers		

* MeSH terms are indicated with a “/” after the search term. Programmes, setting and outcome search terms were combined with “AND” in each database.

**Table 3 ijerph-19-09202-t003:** Inclusion/Exclusion Criteria.

Order	Criteria	Inclusion Criteria	Exclusion Criteria
1	Language		Paper not published in English
2	Date of Publication	Published between 1 January 2011–31 July 2021	
3	Access to Publication	Full Paper Access via UCL/Online	Paper not fully available online
4	Type of Paper		Papers without an Abstract
5	Publication Type	Original Studies Peer Reviewed Studies	Systematic Reviews Editorials Dissertations Not Fully Published Papers
6	Setting	Conducted in the UK or an OECD Country Reporting an intervention that was conducted in a workplace	Reporting interventions in health sector workplaces (e.g., a hospital)
7	Outcome Measured	Quantitative data on workplace stress or anxiety (burnout, perceived stress, measures of cortisol levels, etc.)	Change in outcome level not reported
8	Population Group	Reporting an intervention which provides data on its effects on older workers in the workforce	Data reported with no desegregation by workers age or no evidence that included workers would be considered as “older”
9	Study Design	Experimental Designs Randomised Controlled Trials Non-Randomised Trials Before and After Studies	Qualitative papers (i.e., interview, focus group or ethnographic studies reporting experience of impressions)
10	Study Aim	Where at least one of the objectives of the intervention or programme is to reduce workplace stress	

**Table 5 ijerph-19-09202-t005:** Results from the Risk of Bias Critical Appraisal.

Study (Author, Year)	Domain 1 (Randomisation Process)	Domain 2 (Deviations from intended Interventions)	Domain 3 (Missing Outcome Data)	Domain 4 (Measurement of Outcome)	Domain 5 (Selection of the Reported Results)	Domain 6 (Overall Bias)
Hughes, 2011 [17]	2	1	2	2	1	2
Malarkey, 2013 [18]	2	1	1	1	1	1
Aikens, 2014 [19]	2	2	1	1	1	1
Cook, 2015 [20]	2	1	2	2	1	2
Largo-Wight, 2017 [21]	2	1	1	2	2	2
Limm, 2011 [22]	2	1	1	2	1	1
Hoeve, 2021 [23]	1	1	1	1	1	1
Ojala, 2019 [24]	2	1	2	2	2	2
Fischetti, 2019 [25]	2	1	1	3	2	2
Calogiuri, 2016 [26]	2	1	2	2	2	2

Key for Table 5: 1 = Low Risk of Bias. 2 = Some Concerns. 3 = High Risk of Bias.

**Table 6 ijerph-19-09202-t006:** Description of Methods and Findings.

**Study** **(Author, Year)**	Intervention Type	Duration	Location	Outcome Measurement Method	Data Collection Type	Findings	Intervention Deemed Effective? ^1^
Low risk of bias							
Malarkey, 2013 [18]	Psychological	8 Weeks	Office	Perceived Stress Scale Questionnaire	Self-report	No significant differences were seen at follow-up	No
				Cortisol Levels	Blood test & Saliva sample	No significant changes were noted	
Aikens, 2014 [19]	Psychological	7 Weeks	Online	Perceived Stress Scale Questionnaire	Self-report	23.1% decline in perceived stress	Yes
Limm, 2011 [22]	Psychological	8 Months	Office	Stress Reactivity Scale (SRS)	Self-report	The reduction in SR in intervention group (from 54.5 to 50.2) was significantly higher than in the control group (from 54.5 to 52.7)	Mixed
				Salivary Cortisol Levels	Saliva samples	No effect observed	
Hoeve, 2021 [23]	Psychological	6 Weeks	Office	Depression, anxiety, and stress scale (DASS)	Self-report	General stress score decreased from a group mean of 1.05 to 0.58	Yes
				Police Stress Questionnaire (PSQ-Op)	Self-report	Occupational Stress scores decreased from a group of 3.17 to 2.84	
Some bias concerns							
Hughes, 2011 [17]	Educational & Psychological	12 Months	Office	Perceived Stress Scale Questionnaire	Self-report	No quantitative data reported ^2^	No
Cook, 2015 [20]	Physical & Psychological	3 Months	Online	Symptoms of Distress Likert scale questionnaire	Self-report	No difference between groups	No
				Coping with Stress questionnaire	Self-report	No differences between groups on coping with stress	
Largo-Wight, 2017 [21]	Psychological	4 Weeks	Outside	Perceived Stress Scale Questionnaire	Self-report	Mean PSSQ score decreased from 62.3 to 61.2 in intervention group, compared to 66.2 to 64.2 decrease from control group	Yes
Ojala, 2019 [24]	Psychological	9 Months	Implied to be in Office	Bergen Burnout Inventory (BBI)	Self-report	Total BBI decreased from 36.9 to 33.9 in intervention group, at follow up	Yes
				Utrecht Work Engagement Scale (UWES)	Self-report	UWES increased from 4.3 to 4.5 in intervention group, at follow-up	
Fischetti, 2019 [25]	Physical & Education	8 Weeks	Office	Occupational Stress Indicator	Self-report	Scores for Job as a source of stress decreased from 30.7 to 25.2	Yes
				Short-Form 12 Questionnaire	Self-report	Increases in scores with significant changes from pre- to post-intervention (48.2 to 53.4)	
Calogiuri, 2016 [26]	Physical	2 Weeks	Outside	Physical Activity Affective Scale Questionnaire	Self-report	Higher ratings for PAAS, in relation to Positive Affect in the intervention group	Mixed
				Cortisol Awakening Response (CAR)	Blood Test	No significant differences between groups	

^1^ For the Intervention Deemed Effective Column: Yes = the paper’s authors classify the intervention as effectiveness and/or successful in the text of the paper. Mixed = the paper has unclear conclusions regarding the effective of the intervention report. No = the intervention was not deemed as being effective by the authors of the paper. ^2^ In Hughes et al.’s paper there are no quantitative data for stress levels in the main text. However, a supplementary paper with that data was said to be available via contact with the Journal or Author. After contacting both, no response was received.

## Data Availability

Not applicable.

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
