# Peer review of "Workplace Interventions to Reduce Occupational Stress for Older Workers: A Systematic Review"

_ijerph, 2022, doi:10.3390/ijerph19159202_

Round 1

Reviewer 1 Report

The theme of the work is relevant. There are previous publications on the topic, but in other populations. However, some points need improvement.

1.     I do not understand why the "'s" after the "et al."

2.     PRISMA is a guide to writing systematic reviews, not a protocol. Text adjustments need to make this clear.

3.     In table 1, what would be “usual practice”? That needs to be very well defined.

4. Could you explain better why table 2 has three columns of search terms?

5. Your strategy is missing information. New research is added every day to these databases. In period was your search done? Where were these terms searched? Titles? Abstracts? Full paper?

6.     OECD country and UK. Please only use your acronym after you have previously described it.

7.     In table 1, your population is “Older, office workers, in the UK”, but in line 112 you say that “The 111 papers had to have been conducted in an OECD country”. Although it serves as a comparison with UK workers, the information has to be compatible.

8.     Your table 3 needs improvements. The exclusion criteria are not the opposite of the inclusion criteria. Exclusion criteria are to exclude papers that have already been included. If the inclusion criterion is “Paper published in English”, papers that are not in English cannot be an exclusion criterion. You can only delete what has already been included. That is true for other criteria in this table.

9.     Again, criterion 6 in table 2 does not agree with the population in table 1.

10. Line 128 and 138 - DS and JS. Make it clear that this is a reviewer.

11. The automation tool used in the paper was Rob 2.0. This tool assesses the risk of bias. In Figure 1, you report that you excluded papers “marked as ineligible by automation tools”. First, the bias assessment did not enter the description of its exclusion criteria. Secondly, papers should not be excluded from a review because of bias. These biases need to be analyzed and weighted, but not excluded. That, in my view, is a major methodological problem because it excludes papers that could be important for your analysis.

12. In table 4, what is RCT? You use too many acronyms without describing them first.

13. In table 1, your population is “office workers”, and in table 4, you include papers with police officers. Police officers would only qualify as office workers if the study were carried out exclusively with police officers restricted to administrative activities.

14. In table 1, the population is from the United Kingdom (UK), but in table 4, there are no studies developed in the UK. In line 111, you report that “The papers had to have been conducted in an OECD country, to allow for comparison with the United Kingdom”, but there is no paper carried out in the UK.

15. There is no comparison between elderly and non-elderly populations! In table 6, when talking about interventions, you do not divide the groups, and you do not make this comparison throughout the text. In line 238, you say “Seven out of the ten papers reported some effectiveness in reducing self-reported stress in older workers as a result of interventions”, but in line 154 you say “Five papers had a specific focus on older workers”.

16. At the end of the paper, I cannot identify which intervention could be more efficient in this population group. I cannot see a comparison between countries, a comparison between the elderly and the non-elderly, and between intervention and “usual practice”. Unfortunately, for these reasons, their conclusions are not supported by their results.

Reviewer 2 Report

The article is of great relevance and is overall well-written and well-presented and easy to go through. Nevertheless, there are some minor points that are not totally clear in the methods, and it would increase clarity addressing them. I would also recommend the authors to increase interpretation of the results, which right now I think is the major change needed. While on one hand I understand that is difficult interpreting an area that is quite blurry in its definition and not many studies are performed, on the other hand making an extra step on the interpretation of the findings it would not just strengthen the review, but it will provide even a stronger point that not much is done in this area, and what is done is often only short term.

INTRODUCTION

1.     Line 40 the authors mention that older workers are found to be at increased risk of fatal occupational injuries. But the citation only partially supports this statement since only 50% of the articles in the cited review find older workers at increased risk of fatal workplace injuries, while the remaining 50% does not. Thus wouldn´t it be more correct to say that the literature has so far provided contradicting results?

2.     On the same line, it is important that when considering occupational injuries several mechanisms may be playing a role in why older workers may look as if they have increased risk of injuries. One is that it is important to consider time at risk for the injuries: older workers generally work full-time while younger people may have short term contracts and sometimes part-time. Thus, these would put young workers at less time at risk for eventually having an occupational injury. Moreover, a recent manuscript has found that under-reporting of occupational injuries is higher among younger workers because of many reasons (fear of reappraisal, less workplace rights etc.) and this would mean that in terms of incidence that would result in an undercount of injuries among the younger working population. I would thus recommend the authors to be careful when citing literature and that different mechanisms may lie behind differences in health outcomes among young and older workers.

3.     At line 64 and 68 the authors point out at the fact that much of the existing literature focuses on white collar workers as well as younger workers. But isn´t this review doing partially the same? From the key words (in the method section) it seems as if this review includes only office workers? If so, what is the rationale for this? Why not -as mentioned in the introduction- include also non office workers?

4.     At line 78 the authors mention that this review is relevant for policy makers and employers. I believe this study has the potential to provide evidence also to union representatives as well as workers themselves, so that they could eventually vouche for successful intervention.

METHODS

5.     In table 1, is the comment mentioned above, why do the authors decide to restrict on office workers when from the introduction the reader is led to think that this review is going to include all workers?

6.     In table 2, why are terms different across databases? While sometimes the way the search is built is different across databases, wouldn´t key terms remain the same?

7.     At Line 120, the author mention “If the mean age of participants in a paper were within 15 years of normal pension age, it was concluded that older workers were included in this intervention”. It is not clear how this cutoff was chosen by the authors so perhaps some clarification is needed. Also, if 63 is the normal pension age, if you deduct 15 years, a worker aged 48 would be considered old. But can we think of someone as a old workers if the person is working an office in the same way of some 48 but working in a quite physical deteriorating occupation? It is not clear throughout the manuscript.

8.     At line 148 in the method, there is no explanation on why the authors decide to perform such critical appraisal. But why do the author decide to base  the assessment upon what was written in each paper? Since only 10 articles are included, wouldn´t we benefit for a deeper assessment where the authors provide their own assessment perhaps on top of that of the included articles?

9.     Along the same line, since no statistical calculations were performed to assess the manuscripts and the authors mention that only a small number of articles were found, why not including qualitative studies in the review where perhaps workers are asked whether an intervention was perceived as effective?

RESULTS AND STUDY FINDINGS

10.  Underneath table 5, the authors describe how risk of bias was assessed. I recommend the authors to mention this in the method and explain what low, some and high risk mean to the authors.

11.  At line 2018 I suggest citing which articles are the authors referring to when saying that there was improvement in the stress level in the same way done at line 222.

12.   

DISCUSSION

13.  At the beginning of the main findings (line 238) I suggest specifying that most were short term interventions.

14.  It is not clear in the discussion how do the author interpret that those that assessed the intervention as effective were mostly short interventions (only one lasting 9 months assessed themselves as effective).

15.  It is not mentioned in the discussion how do the author interpret the fact that the among the long-lasting interventions, 2 out of 3 found mixed effects or no effects.

16.  The author rightfully mention the ambiguity in defining old age, but they do not interpret how a fairly “young” working population (the 40 to 48) may actually affect the results found. Could it this be the reason why they 7 out of 10 were successful interventions? Or was it more because they are short-term so the effectiveness is debatable?

17.  At line 276, the author give a nice interpretation on age among policeman. But I do not totally follow the reasoning from line 279. I would imagine, that because of a natural process of growing within an occupation, perhaps police officers around the age of 40 reach a certain seniority and thus work more on “desk” tasks and cover more managerial positions and since this review includes mainly office workers, then they will consequently have less risk of injuries.

18.  Much is missing in the interpretation section. What is the authors interpretation on the fact that 7 out of 10 assess their intervention as successful even if most are short term? How do the author think around the fact that the long interventions showed mixed or no results? And having restricted to office, workers, how generalizable are these results to other workers? Are there actually other interventions performed on non-office workers?

19.  Line 298 to 306 seems more fitting the other section, since it is not interpretation.

Reviewer 3 Report

Thank you for the opportunity to review this manuscript. Overall, this systematic review is interesting, well-written, and appropriate for the journal. Please see my detailed comments below:

Introduction

This section was well-written and the topic was described well. The gap in the literature was clearly identified.

Methods

The methods are sound, overall. My questions are limited to two main points: 

-Why were qualitative studies excluded? It's possible to do a mixed-methods systematic review, and these studies may have provided interesting insights. It's fine that they weren't included, but a more justified reasoning should be provided.

-A potential limitation is the exclusion of non-English publications. It is becoming increasingly common to use free, online translation software for non-English studies. Given this, it would be helpful to know how many articles in the search were not included because they were not in English? If the number is significant, it may be beneficial to add this as a limitation.

Results:

The results are clearly presented and discussed.

Discussion:

The discussion is well-written and includes relevant implications.

Conclusion:

These conclusions are supported by the results.

Round 2

Reviewer 1 Report

Congratulations, many questions about the work became clearer. But I still have some questions.

1. You need to identify which intervention you are using as a control. It is not yet clear. What is “current, normal workplace practices”? You can compare with the absence of intervention, but if there is any intervention as a buyer, it is necessary to say which ones. Without it, your results and conclusions are dubious. The core of their study is to compare interventions!

2. You say “United Kingdom or Organization for Economic Co-Operation and Development (OECD) country”, but the United Kingdom is part of the OECD. The information is redundant. Just make writing corrections.

3. Table 3 still has many exclusion criteria that could not be exclusion criteria. If an article has not been included, it cannot be excluded. Think about this when reviewing the table.
